genomics, ecology, evolution

palaeogenomes, *sed*aDNA, integrative, ecosystem shifts, extinction

**Authors for correspondence:**
Anders Götherström
e-mail: anders.gotherstrom@arklab.su.se
Love Dalén
e-mail: love.dalen@nrm.se
Peter D. Heintzman
e-mail: peter.d.heintzman@uit.no

# Integrating multi-taxon palaeogenomes and sedimentary ancient DNA to study past ecosystem dynamics

Nicolas Dussex[1,2,3], Nora Bergfeldt[1,2,3], Violeta de Anca Prado[1],
Marianne Dehasque[1,2,3], David Díez-del-Molino[1,2], Erik Ersmark[1,4],
Foteini Kanellidou[1], Petter Larsson[1,4], Špela Lemež[1], Edana Lord[1,2,3],
Emilio Mármol-Sánchez[1,5], Ioana N. Meleg[1,3,6,7], Johannes Måsviken[1,2,3],
Thijessen Naidoo[1,4,8], Jovanka Studerus[1], Mário Vicente[1,4], Johanna von
Seth[1,2,3], Anders Götherström[1,4], Love Dalén[1,2,3] and Peter D. Heintzman[9]

[1]Centre for Palaeogenetics, Svante Arrhenius väg 20C, 10691 Stockholm, Sweden
[2]Department of Zoology, Stockholm University, Stockholm, Sweden
[3]Department of Bioinformatics and Genetics, Swedish Museum of Natural History, Stockholm, Sweden
[4]Department of Archaeology and Classical Studies, Stockholm University, Stockholm, Sweden
[5]Science for Life Laboratory, Department of Molecular Biosciences, The Wenner-Gren Institute, Stockholm University, Stockholm, Sweden
[6]'Emil Racoviţă' Institute of Speleology of the Romanian Academy, Calea 13 Septembrie, nr. 13, 050711, Sector 5, Bucharest, Romania
[7]Emil. G. Racoviţă Institute, Babeş-Bolyai University, Clinicilor 5–7, 400006 Cluj-Napoca, Romania
[8]Ancient DNA Unit, SciLifeLab, Stockholm and Uppsala, Sweden
[9]The Arctic University Museum of Norway, The Arctic University of Norway, 9037 Tromsø, Norway

ND, 0000-0002-9179-8593; NB, 0000-0003-2767-8156; VdAP, 0000-0003-1845-509X;
MD, 0000-0002-4640-8306; DD-d-M, 0000-0002-9701-5940; EE, 0000-0003-4186-7498;
EL, 0000-0002-4717-1988; EM-S, 0000-0002-4393-1740; INM, 0000-0002-0836-4971;
JM, 0000-0003-2660-7081; MV, 0000-0002-9122-4530; JvS, 0000-0002-1324-7489;
AG, 0000-0001-8579-1304; LD, 0000-0001-8270-7613; PDH, 0000-0002-6449-0219

Ancient DNA (aDNA) has played a major role in our understanding of the past. Important advances in the sequencing and analysis of aDNA from a range of organisms have enabled a detailed understanding of processes such as past demography, introgression, domestication, adaptation and speciation. However, to date and with the notable exception of microbiomes and sediments, most aDNA studies have focused on single taxa or taxonomic groups, making the study of changes at the community level challenging. This is rather surprising because current sequencing and analytical approaches allow us to obtain and analyse aDNA from multiple source materials. When combined, these data can enable the simultaneous study of multiple taxa through space and time, and could thus provide a more comprehensive understanding of ecosystem-wide changes. It is therefore timely to develop an integrative approach to aDNA studies by combining data from multiple taxa and substrates. In this review, we discuss the various applications, associated challenges and future prospects of such an approach.

## 1. Introduction

The development of ancient DNA (aDNA) as a scientific tool can be divided into three phases. First came the realization that DNA could be recovered from ancient remains and thus offer a temporal dimension to genetic analyses that modern data alone cannot provide [1]. This was followed by a period when most studies were focused on recovering DNA from different taxa and placing them into a phylogenetic context. Technical advances during this period, most

notably the development of the PCR method [2] and use of silica for DNA extractions [3], paved the way for studies on the genetic relationships between extinct species and their extant relatives (e.g. flightless ratites [4]). However, several of these early studies, such as those on Cretaceous remains [5], are today considered the result of contamination, and therefore erroneous (e.g. [6]).

The second phase was catalysed by a series of seminal studies that made use of population-level datasets of short mitochondrial DNA sequences to investigate within-species demographic histories (e.g. [7]) as well as the origin of domestic species (e.g. [8]). These studies revealed a general pattern of dynamic history during the Late Quaternary, often characterized by population replacements and losses of genetic diversity. During this phase, it was also demonstrated that short barcode sequences recovered from ancient sediments or faeces could be used to examine the composition of prehistoric plant and animal communities or the diet of ancient taxa [9,10].

The third phase was initiated by the emergence of new DNA sequencing technologies and their application to aDNA [11]. This enabled aDNA to mature into a tool useful for a broad spectrum of scientific disciplines. The development of high-throughput sequencing methods also enabled the emergence of robust studies of ancient pathogens [12] and their importance for human prehistory [13], microbiomes [14] as well as the high-resolution reconstruction of past ecological communities from sedimentary aDNA (e.g. [15–17]). The first publications of complete prehistoric human and Neanderthal genomes [18,19] opened the floodgates for studies using ancient genomes (palaeogenomics), especially to trace human gene flow across continents [20]. The past decade has also seen an increase in the use of palaeogenomics to study population change, gene flow and extinction dynamics in wild and domestic animals [21,22]. Overall, the recent analyses of large-scale palaeogenomic datasets have been highly successful in investigating species-specific population histories.

## 2. Using palaeogenomics to investigate single-species histories

Palaeogenomics has been used to investigate species' histories, including changes in population size and gene flow. Bayesian coalescent methods have been used to reconstruct past changes in female effective population size ($N_e$) from mitochondrial genomic data [23], whereas sequentially Markovian coalescent (SMC) methods have made demographic analyses from single ancient nuclear genomes routine (e.g. Neanderthals [24]; woolly mammoths [21]).

The increasing availability of dated genomes from modern and ancient human populations [20] and domesticated species (e.g. horses [22]; canids [25]) has allowed for the inference of ancestral relationships between populations using ordination methods, such as principal component analysis (PCA) or, more recently, factor analyses (FA) [26], the latter of which properly accounts for sample age and temporal drift.

The generation of ancient genomic data has also spurred the development of methods to detect admixture between closely related species, including when hybridizing species are extinct [27]. For example, the sequencing of the first Neanderthal genome indicated that non-African modern human genomes comprise approximately 2% Neanderthal DNA [19]. These methods are now routinely used in palaeogenomics studies and have also contributed to a recent surge in studies on hybridization in a wide variety of modern taxa including insects, plants, mammals, birds and fish, and indicate that ancient admixture between related populations and species was commonplace (reviewed in [28]).

Selection and domestication studies have also benefited from the inclusion of palaeogenomic data. The temporal dimension provided by aDNA can allow for the study of changes in allele frequencies 'in real time' [29]. For instance, palaeogenomic data from Early Neolithic and Bronze Age Eurasian humans enabled a deeper understanding of the genetic basis of lactase persistence [30]. Palaeogenomics is beginning to provide valuable contributions to the study of natural selection in extinct taxa and has been used to investigate genetic changes associated with adaptations to cold climates [23,31], predatory lifestyle, behaviour and morphology [32] or the roles of natural selection and genomic diversity in extinction [33]. Finally, the inclusion of palaeogenomic data has been also necessary for studies on domestic species in which wild or past domestic lineages are currently extinct, such as horses [34].

## 3. Sedimentary ancient DNA adds another dimension

aDNA recovered directly from lake, cave, permafrost, archaeological or other environmental sediments (sedaDNA) is a rapidly evolving tool that holds much promise. As sediments, and the aDNA incorporated within them, are often deposited gradually and continuously over time, they can be used to reconstruct past ecological communities at fine taxonomic and temporal resolution and provide local first and last appearance dates (FADs, LADs) for taxa independent of the completeness of the body fossil record (e.g. [35,36]). Similarly, the recovery of aDNA from associated unidentifiable bulk fossil fragments can supplement sedaDNA data extracted directly from sediment (e.g. [37]). Integration of these data can, therefore, provide a detailed record of community changes that occurred across times of arrival and extinction of keystone taxa, such as mammalian herbivores.

The first reported recovery of sedaDNA was the bacterial profiling of lake sediment [38], with the first evidence for plant and animal sedaDNA reported from caves and permafrost [9,39]. Subsequently, the majority of studies have used PCR-based DNA metabarcoding methods to amplify sedaDNA molecules of interest from individual broad taxonomic groups (e.g. plants or mammals [40–42]). Advanced methods that sequence entire sedaDNA molecules, and thereby allow for aDNA damage authentication (see also §5a), have only recently been applied. These methods include shotgun metagenomics, whereby any molecules in the sedaDNA mixture are randomly sequenced (e.g. [15,36,43–46]), and target enrichment, in which sedaDNA molecules of interest are selectively enriched prior to sequencing (e.g. barcode or mitochondrial loci [16,47–49]). Detailed descriptions of these methods applied to sedaDNA have been recently reviewed elsewhere [50,51].

The recovery, analysis and interpretation of sedaDNA poses significant challenges, in part due to the complex

mixture of ancient ecosystem DNA present in a sediment sample. Nonetheless, progress is rapidly being made to address these issues, which we detail in §5a, and many valuable contributions from *sed*aDNA have already been made. For example, detailed plant community reconstructions now exist for sites from multiple regions (e.g. [17,52]) and interglacial periods [41], hominin and human *sed*aDNA has been recovered (e.g. [47,48,53]), LADs have been refined for extinct megafauna (e.g. [36]), and FADs have been established for taxa arriving in a variety of contexts, from newly deglaciated landscapes [15] to island invasions [35]. Several studies have integrated *sed*aDNA findings with other palaeoecological proxy data to provide additional validation and/or contextualization (e.g. [36,52]). However, multi-site comparative *sed*aDNA studies (e.g. [42,54]) are still rare.

With the application of shotgun metagenomics and target enrichment approaches, it is now possible to recover haplotypic and genomic information directly from *sed*aDNA [43–45,47–49], which enables the exploration of population-level changes and has the potential to detect the arrival or disappearance of alleles and lineages in a region, as recently showcased for Neanderthals from a cave in Spain [48]. This expansion of *sed*aDNA into environmental palaeogenomics, together with the integration of *sed*aDNA and traditional palaeogenomic data derived from body fossils [55], will open up new approaches to understanding past biodiversity changes that are inaccessible with other palaeoecological proxies.

# 4. Integrating data from humans, animals and sediments

Genetic studies on modern-day samples have successfully integrated genomic and/or epigenomic data from multiple unrelated taxa (the multi-taxon approach) to address a range of questions in evolutionary biology (e.g. [56,57]), such as inferring the distribution of pathogens linked to early human migrations [58] using comparative phylogeographic approaches.

Generating multi-taxon datasets in palaeogenetics has been limited by sparse fossil records, the degraded nature of aDNA, contamination with modern DNA, sequencing costs and computational resources [59,60]. Although palaeogenetic data from multiple taxa have been used to contrast demographic histories (e.g. [61,62]), recent genomic studies have inferred the genetic ancestries and histories of multiple mammalian taxa from a single Pleistocene cave *sed*aDNA sample [44,45]. Another study inferred a clear parallel between dog and human lineage diversification by overlaying their population histories [25]. To our knowledge, this is the first study that used a multi-taxon approach by quantitatively coanalysing palaeogenomes of two coeval and cospatial species and thus paves the way towards a multi-taxon approach in aDNA studies.

In spite of the numerous technological and computational challenges in palaeogenomics, the increasing number of ancient genomes from wider geographical and deeper time scales (e.g. [27]) will enable the genomic history of numerous species to be unravelled, whereas *sed*aDNA will allow for direct evidence of the timing and extent of associated past ecological changes. However, appropriate statistical frameworks to quantitatively coanalyse intra-taxon/inter-taxa genomic patterns across time and space are required to

overcome the inherent heterogeneity in such datasets (i.e. species from different spatio-temporal contexts) that may bias data interpretation. For instance, integrating distributional, demographic and coalescent modelling (iDDC) with approximate Bayesian computation (ABC) has been proposed as a methodological transition to coanalyse species datasets under biologically informed hypotheses [63].

We here propose that recent advances in the generation and analysis of high-throughput sequencing data provide new opportunities to formally integrate multi-taxon knowledge from palaeogenomic and *sed*aDNA data into a cohesive picture of human–animal–environment interactions in the past (figure 1). In the following subsections, we discuss how aDNA data integration could provide new insights into the interaction between humans, wildlife and domesticated animals, and changes in their immediate environment. We also give an overview of the technical challenges and future prospects for the promising development of integrative approaches to build comprehensive and coherent datasets within a holistic aDNA evolutionary perspective.

## (a) Consequences of human arrival on wildlife
Thanks to their ability to adapt to a wide range of climatic and geographical conditions, humans have impacted ecosystems globally through hunting, domestication, sedentarization, and land and resource exploitation. Anatomically modern humans (AMHs) originated in Africa at least 200 000 years before present (ka BP), and expanded outside the continent within the past 100 ka BP [64], reaching North America by at least 16 ka BP and Polynesia around 1.0–0.7 ka BP [65]. Furthermore, changes in human technology that allowed for more efficient hunting or to target a specific species, such as the development of hunting tools used by Clovis hunters, are thought to have accelerated demographic declines in wild populations [66]. Similarly, the dispersal of Neolithic farmers from the Fertile Crescent across Europe and the introduction of their agricultural practises and domestic livestock from approximately 11 ka BP [67] followed by their sedentarization may have induced important changes in the environment [68].

It has been suggested that human arrivals had significant impacts on previously unoccupied areas and were characterized by a number of extinctions as a result of overhunting and/or the introduction of non-native predators, particularly in island ecosystems [69,70]. In order to further elucidate the effects of human arrival on taxa, it is essential to refine the timing of first human presence in different regions. *sed*aDNA is a potentially valuable tool for detecting human FADs when macrofossil remains are sparse. Fine-scale information regarding human arrival and migrations could then be used to correlate the timing of human arrival with demographic declines in fauna inferred from palaeogenomic data. Multi-taxon demographic reconstructions using palaeogenomes can, for instance, be used to establish whether native taxa were impacted synchronously by human arrival (figure 2), and whether their extinction pattern is better explained by differences in life-history traits or body mass (e.g. megafauna [71]). Such information may also help elucidate whether wildlife populations that may have already been declining due to external factors (e.g. climate change) were more vulnerable to the arrival of human populations.

Proc. R. Soc. B 288: 20211252

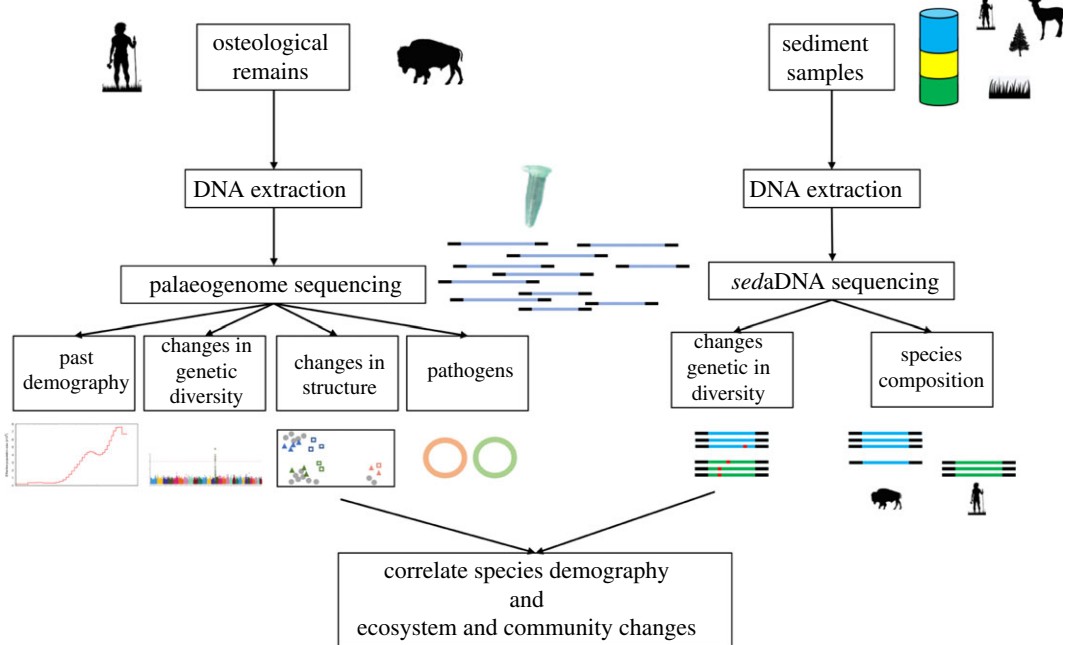

**Figure 1.** Workflow for the integration of multi-taxon palaeogenomes and *sed*aDNA. Silhouettes are from PhyloPic.org. (Online version in colour.)

Similarly, *sed*aDNA could be used to test for ecosystem changes and examine the impact of human arrival on the abundance of another species in real time, in an approach similar to Gelabert *et al.* [45]. For example, early human populations may have competed with cave-dwelling species for shelter (e.g. [72]). Here, *sed*aDNA could be used to test whether humans and other cave fauna co-occur or are mutually exclusive.

## (b) Correlating human and animal demographies

An important question that remains to be addressed is whether there are tipping points of human population densities that could trigger significant declines in the demography of prey species. For instance, while human arrival in northeastern Siberia probably did not impact woolly rhinoceros demography, subsequent changes in human population density, which are currently unknown, may have had such an impact [23]. Thus, future work needs to focus not only on the effect of arrival, but also on correlating human and wildlife demographic trajectories thereafter.

Comparative analyses of palaeogenomic data could allow for this question to be tested by examining the impacts of human interference on species demography (e.g. using SMC; figure 2). Furthermore, larger multi-taxon datasets would enable testing of correlations among species, using different estimates of genetic diversity (e.g. $F_{ST}$, inbreeding), and provide evidence for anthropogenic impacts on wildlife. For example, palaeogenomic data indicate that human demographic events are correlated with dog population history and that the expansion of steppe pastoralists in Eurasia caused a complete replacement of European domesticated dog genetic diversity [25]. Such an approach could also be used to test to what extent local hunting pressures impacted the population dynamics of the extinct Baltic Harp seal [73].

Ideally, a combination of *sed*aDNA and palaeogenomes from fossil remains would allow examination of inter-taxon interactions across entire ecosystems. For instance, human dispersal into Australia and North America may have led to megafaunal extinctions, declines and range shifts which could be examined in time and space with these types of data in combination with modelling of interactions among taxa and changes in $N_e$ through time (e.g. predator–prey models) [74]. Because megafaunal extinctions are often complex and multifactorial, a multi-taxon palaeogenomics approach will be especially valuable for assessment of the respective roles of human and non-human environmental changes in species extinction. In this regard, we stress that ethical considerations, engagement with indigenous communities, as well as careful interpretation of the narrative stemming from these discoveries, will be essential to avoid any potential stigmatisation of indigenous peoples [75].

## (c) Cascading effects of species extinctions

Species extinctions can have cascading effects on the physical and trophic structure of ecosystems, as well as the diversity and evolution of species. These effects could be examined by comparing demographic trajectories of several species simultaneously from multiple aDNA sources (e.g. *sed*aDNA, subfossil remains).

The extinction of ecologically important species can cause ecosystem state shifts. For example, extinct megaherbivores such as woolly mammoth and woolly rhinoceros are thought to have maintained a mosaic of open and shrub habitats characterized by high plant diversity [76], through grazing, soil fertilization and seed dispersal [77]. Consequently, the extinction of these large herbivores may have led to a shift towards dense and closed vegetation, a reduction in diversity and the extinction of species that had coevolved with these dispersers [78]. Moreover, large-bodied herbivores play an important role in maintaining connectivity between habitat patches through seed and nutrient dispersal [79]. *sed*aDNA could help identify changes in plant and invertebrate diversity, whereas genome-wide palaeogenetic data recovered from remains could allow for the testing of changes in connectivity (i.e. gene flow) between patches, thereby indicating whether such changes coincided with megaherbivore extinctions.

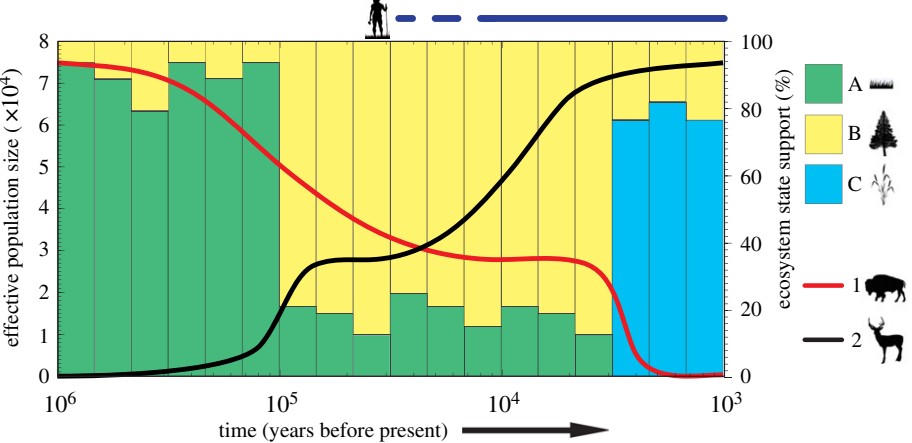

**Figure 2.** Conceptual illustration of joint analysis of aDNA from multiple substrates. Red and black lines depict hypothetical changes in effective population size ($N_e$) inferred from the palaeogenomes of two distinct taxa (e.g. from a PSMC analysis). Filled bar colours represent three different ecosystem states (A–C) derived from sedaDNA. In this example, there are two distinct ecosystem state shifts. The $N_e$ of taxon 1 is in decline prior to the ecosystem state shift from A to B. Its $N_e$ remains stable after humans appear (dark blue line) but crashes during the shift from ecosystem state B to C. By contrast, the $N_e$ of taxon 2 rapidly increases during the first ecosystem state shift (A–B), and again increases after the appearance of humans. The $N_e$ of taxon 2 is unaffected by the second ecosystem state shift (B–C). Silhouettes are from PhyloPic.org. (Online version in colour.)

Single-species extinctions can also affect trophic interactions by triggering a number of secondary extinctions. For instance, the extinction of prey species can induce the disappearance of its predator [80]. Conversely, the extinction of an apex predator can lead to mesopredator release via reduced mortality and competition [81]. Furthermore, because apex predators regulate herbivore populations [82], the extinction of these predators could lead to changes in herbivore abundance, thereby altering trophic cascades and habitat structure and vegetation. A combination of sedaDNA and comparisons of population trajectories from subfossil remains would enable the testing of secondary extinctions and mesopredator release hypotheses (figure 2).

Another important consequence of species extinction is that it can trigger adaptive evolution in other species. For example, the extinction of carnivores could trigger a change in body size of herbivore prey species, similar to what has been proposed for the evolution of island herbivores in the absence of predators [83]. Conversely, the extinction of large prey species may have caused body size reduction in predators and scavengers [84]. Examining temporal changes in adaptive variation from scavenger and predator remains based on demographic reconstruction of its extinct prey could thus be used to test whether a reduction in body size has a genetic basis and whether it coincides with the extinction of their prey.

Finally, because many megafaunal species represented important food, building material, tool and artefactual resources for humans (e.g. [85]), thereby contributing to shaping human cultures [86], megafaunal extinction may have triggered migrations and perhaps even local extinction of human populations, as well as dietary shifts [87]. Decline or extinction of important prey species may even have contributed to cultural shifts towards new hunting strategies and subsequently domestication [88]. Using multiple aDNA sources, it should be possible to test whether the extinction of specific megafauna triggered changes in human demography, culture (e.g. changes in diet) and/or population turnovers. Moreover, examining extinction-driven cascading effects constitutes a 'natural experiment' that can be used to test whether particular ecosystems are under bottom-up or top-down control, a question that is still heavily debated in ecology.

### (d) Impact of domestic species on wild animals

As human populations expanded into new regions and impacted wild populations through hunting and habitat change, domesticated species that they brought with them also had a significant impact on these new ecosystems. For instance, since dingos were used to assist human hunting of small and large prey, their introduction in Australia probably contributed to the extinction of the thylacine and the Tasmanian devil on the mainland [89,90]. Similarly, the use of hunting dogs on other continents may have led to a higher hunting success and increased pressure on ungulate populations (e.g. ibex, gazelle [91]). Conversely, the replacement of hunter–gatherer populations by Neolithic farmers bringing domesticated taxa with them may have led to a relaxation of hunting pressure on wildlife. A combination of sedaDNA with demographic reconstructions for native wild species would thus enable testing of whether these declines or extinctions continued or stopped with the introduction of domesticated animals and farming cultures.

Another direct consequence of the introduction of domestic species to new ecosystems is introgression between domestic taxa and their wild counterparts both in modern (e.g. wild boar [92]) and ancient times (e.g. horses [22]; wolves [25]). Comparing genomes from a wild population and the domesticated species prior to and after the arrival of the latter could help resolve whether and when introgression occurred.

The introduction of domesticated species can also have indirect impacts on native fauna, with, for instance, the spread of both parasites and pathogenic microbes from domestic dogs to several wild canid species [93]. It is thus likely that similar transfers of diseases and pathogens occurred upon first contact between domesticated and wild animals. Consequently, comparing pathogens found in domesticated and related or unrelated wild taxa using a metagenomics approach could be used to test the hypothesis that the earliest domesticated arrivals were vectors of diseases into wild populations.

### (e) Human-driven landscape change

The dispersal and subsequent sedentarization of human populations had a severe impact on landscapes and

ecosystems. These effects were most significant during the Neolithic transition, following the shift from hunter–gatherer to farming cultures (e.g. [94]). This shift entailed a steady decline and fragmentation of forested areas through land clearing as well as a profound alteration of aquatic ecosystems through irrigation and wetland draining [40], which likely had important effects on animal species [95], plant communities [94] and associated trophic networks.

Integrating aDNA data from sediments, bones, coprolites and other archaeological remains with data from more traditional methods (e.g. radiocarbon dating, pollen, macrofossils) could help infer the timing of human arrival and provide a comprehensive understanding of the effect of humans on the landscape. Moreover, because the Neolithic transition occurred at a time when the climate in Europe changed and sea levels were rising [96], this integrative approach could enable to disentangle the roles of human activities and climate change in the transformation of Holocene landscapes.

For instance, deforestation, grazing by domestic animals, and other human impacts in Iceland and Iberia during historical periods led to severe erosion, soil depletion and desertification [97,98]. An integrative approach targeting aDNA from plants, vertebrates and soil microorganisms could help unravel the cascading effects of deforestation and erosion on ecosystems. Furthermore, this approach could indicate whether changes in the genetic diversity of forest species coincided with an increase in human-induced landscape change or hunting. Similarly, combining aDNA from aquatic animal and microorganism remains could elucidate how human alterations of waterways due to irrigation and drainage affected aquatic plant and animal populations. Other prospects for aDNA are to investigate other types of human activities, such as the creation and development of man-made soils (i.e. anthrosols) that occur around the world, as well as to test whether the 'elm decline' approximately 6.3 ka BP [99] was caused by a fungal disease or human overexploitation.

# 5. Technical challenges and future prospects

## (a) Challenges inherent to palaeogenomics and sedaDNA research

aDNA research has rapidly advanced over the past 3 decades and challenges associated with DNA damage and modern contamination [59,60] have since been mitigated to a great extent. Yet, the presence of damaged exogenous DNA such as fragmented and deaminated DNA from bacteria and other non-target organisms may show false similarity to the reference genome used and become erroneously incorporated into the target sequence [59,60], thereby leading to incorrect inferences.

Significant challenges specific to sedaDNA research remain, whereby the sedaDNA composition of a sample is subject to intrinsic biases that need to be considered during analysis and integration with other data. For instance, because DNA preservation is reduced in warm and wet environments compared to dry and cold locations, the comparability of time scales and extents of detection across ecosystems may be limited. Generalizable and scalable approaches will, therefore, need to be developed to ensure robust harmonization of datasets using, for example, data

quality metrics (e.g. [42]). Sampling from multiple comparable locations and using biological replicates and negative controls is essential to ensure proper characterization of a target area and to reduce taxonomic bias and the influence of contamination with modern DNA (e.g. [49,100]). Issues of taxonomic bias and sample heterogeneity are further confounded by a paucity of knowledge on sedaDNA taphonomy (the processes by which DNA is transported from an organism into an environmental archive [101]) and preservation, although experimental studies are beginning to address these unknowns (e.g. [102]). Post-depositional vertical movement of DNA via leaching [103], which could potentially lead to erroneous temporal interpretations, can be assessed by comparing sedaDNA data to other proxy and/or contextual information, although recent results from lake (e.g. [36,104]) and cave systems (e.g. [49]) do not find evidence of leaching. Furthermore, molecular biology protocols can be impacted by variations in substrate composition and the co-extraction of inhibitors, although new specialist methods, such as the cold-spin DNA extraction method [16], are beginning to mitigate these issues.

As the proportion of targeted and/or identified sedaDNA molecules may be very low, it is necessary for contamination to be monitored and sedaDNA assignments to be verified and authenticated, where possible (e.g. [46,105]). Although contamination with modern DNA may be excluded by examining aDNA damage patterns, sources of false positive taxonomic assignment could occur from other aDNA molecules that may be short, with low information content, and/or from genomically conserved regions that are shared across taxa. Although the impact of short aDNA molecules in single taxon palaeogenomics datasets, for example, can be characterized and mitigated (e.g. [27,106]), this is not yet true for multi-taxon sedaDNA mixtures and so new quality control methods will be required to determine and reduce false positive taxonomic assignments. An additional insidious source of taxonomic misassignment is incomplete reference databases, which are often sparsely populated and biased towards human-related taxa (exceptions include some databases used for metabarcoding; see [50] for examples). Although the phylogenetic intersection analysis (PIA) has recently been developed to mitigate taxonomic misassignments caused by incomplete reference databases [107], this approach could be developed further by, for example, incorporating geographical and ecological information to probabilistically determine and refine assignments. However, we caution that these data are also often unknown or incomplete for many taxa. Finally, bioinformatic pipelines, taxonomic classifications [42] and estimates of DNA damage [100] need to be standardized to avoid misrepresentation of species and their incorrect interpretation and association with related palaeogenomes at a shared spatio-temporal scale.

## (b) Issues with integrating data from varied sources

While comparative genomics frameworks have been proposed to obtain a better understanding of evolutionary processes, such as connectivity based on modern data [108], there are still several pitfalls when integrating aDNA from multiple sources. First, palaeogenomics or sedaDNA studies are often limited in sample size and the completeness of datasets, which may constrain models and reduce statistical

power. Yet, integrative models can provide a generalized framework for meta-dimensional analysis [109], either using correlation or Bayesian-based models, and can be adapted to more generalistic assumptions for data integration [109]. Furthermore, unsupervised factor transformation methods or deep learning approaches have recently been widely applied for integrating heterogeneous data [110,111] and might be a good fit for particular complex scenarios.

Second, correlations among species demographies may not necessarily indicate a real impact of one species on another. This represents a significant challenge when testing for anthropogenic impacts on wildlife, for instance, because $N_e$ is often smaller than census size ($N_C$) [112]. Thus, a small $N_e$ does not mean that $N_C$ did not reach the threshold necessary to induce a faunal decline. This challenge is analogous to the disconnect between Y chromosome and mtDNA $N_e$ when comparing male and female demography. Using several time points to estimate relative changes in $N_e$ could, however, circumvent this issue.

Finally, there may be a temporal disconnect between demographic trajectories among species. For instance, because $N_e$ will be reduced more slowly than $N_C$, there may be an observable time-lag in demographic reconstructions. One may thus erroneously exclude or infer a causal link between a change in the $N_e$ of one species and a change in the abundance of another species. Such a disconnect can also be expected between changes in abundance in one species and a genome-wide or phenotypic response in another. For instance, while predator release may induce an increase in body size of the prey, this phenotypic change may take time, which is a function of $N_e$ and generation time, and thus also result in a time-lag.

## (c) Future prospects

Future analytical developments will facilitate the integration of different forms of palaeogenetic data. Multi-taxon evolutionary dynamics may uncover molecular signatures of population differentiation as well as infer shared population histories. Recently, cross-species analyses have focused on how different ecologically connected species in the same ecosystems perceive and adapt to changes in the environment

through time [108]. Whole-genome sequencing of selected individuals from different related populations across specific ecosystems have been used to infer micro- and macro-evolutionary connectivity patterns [108,113]. Extending such a framework to palaeogenomics and sedaDNA studies could add useful information on evolutionary dynamics across species and through time in a given region. Similarly, the analysis of inter-species dynamics using aDNA based on a multispecies coalescent model [111] is becoming more popular. For instance, a multi-taxon application of such coalescent models may allow for the joint inference of demographic and evolutionary parameters such as mutation rate, selection, and population expansions or contractions [114]. The integrated use of palaeogenomic and sedaDNA data would thus become more relevant to disciplines such as ecology, in order to infer whether particular ecosystems are under bottom-up or top-down control and could be incorporated in integrated population models or species distribution models [115,116]. The inclusion of temporal and spatially scaled integrated aDNA will help to improve these models, especially when assessing biodiversity changes or species population trends over time.

Data accessibility. This article does not contain any additional data.

Authors' contributions. L.D., A.G. and P.D.H. conceived the review; all authors drafted the manuscript; N.D., E.E. and P.D.H. drew the figures; N.D. and P.D.H. revised the manuscript. All authors gave final approval for publication and agreed to be held accountable for the work performed therein.

Competing interests. We declare we have no competing interests.

Funding. We acknowledge support from the Carl Tryggers Foundation (Grant CTS 19: 257 to N.D.), the Swedish Research Council (VR; 2017-04647 to L.D.; 2019-00849 to A.G.), FORMAS (2015-676, 2017-00704 and 2018-01640 to L.D.), the Bolin Centre for Climate Research (to E.L.), the Strategic Research Area (SFO) programme of the Swedish Research Council through Stockholm University (to E.M.-S.), the Stockholm University's Paired PhD Student Programme (to N.B.) and the European Union's Horizon 2020 research and innovation programme under the Marie Skłodowska-Curie grant agreement (no. 885088 to I.N.M.).

Acknowledgements. The authors are grateful to the staff at Tovetorp Research Station for their support during the writing of this manuscript. We thank two anonymous reviewers for their helpful comments on the text.

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
