## [Peer Review File · Proceedings of the Royal Society B: Biological Sciences]

Review History

RSPB-2021-1252.R0 (Original submission)

Review form: Reviewer 1

Recommendation

Accept with minor revision (please list in comments)

Scientific importance: Is the manuscript an original and important contribution to its field?

Good

General interest: Is the paper of sufficient general interest?

Excellent

Quality of the paper: Is the overall quality of the paper suitable?

Excellent

Is the length of the paper justified?

Yes

Should the paper be seen by a specialist statistical reviewer?

No

Do you have any concerns about statistical analyses in this paper? If so, please specify them explicitly in your report.

No

It is a condition of publication that authors make their supporting data, code and materials available - either as supplementary material or hosted in an external repository. Please rate, if applicable, the supporting data on the following criteria.

Is it accessible?

N/A

Is it clear?

N/A

Is it adequate?

N/A

Do you have any ethical concerns with this paper?

No

Comments to the Author

This is a timely review and prospectus of ancient DNA research applied to understanding ecosystem dynamics. In particular, the authors focus on paleogenomic and sedimentary metagenomics research and do an excellent job of summing up the major advances in the field as well as the challenges. I have only a few comments that might improve the manuscript. For example, I think that pointing to some of the potential ethical issues associated with sedaDNA is important. In the Americas, Australia and elsewhere, it is important to consult with indigenous stakeholders about research that might take place on their lands and to think carefully about the narrative that is used to explain the results of the research (see <https://doi.org/10.1038/s41559-020-01351-6>). Section b is one place where this could be mentioned, since in lines 199-203, the authors note that it would of interest to investigate the arrival of humans and the link to megafaunal extinction. Megafaunal extinction was likely a complex process where humans may or may not have been important players.....but we need to avoid simplistic narratives leaving the general public with the conclusion that "X people killed off the Y animal(s)", particularly where the descendents of "X people" are politically marginalized and when a more contextualized and complex answer is more likely to be true.

Additional comments:

Line 254: The following sentence could be clearer. "For instance, the introduction of the dingo in Australia likely contributed to the extinction of the thylacine and the Tasmanian devil on the mainland [85], whereas the use of hunting dogs led to a higher hunting success and increased pressure on ungulate populations (e.g., moose [68])." Specifically, I was trying to figure out whether the dingo was a hunting dog (were they?)...and what moose were in Australia.

Review form: Reviewer 2

Recommendation

Accept with minor revision (please list in comments)

Scientific importance: Is the manuscript an original and important contribution to its field?

Good

General interest: Is the paper of sufficient general interest?

Good

Quality of the paper: Is the overall quality of the paper suitable?

Acceptable

Is the length of the paper justified?

Yes

Should the paper be seen by a specialist statistical reviewer?

No

Do you have any concerns about statistical analyses in this paper? If so, please specify them explicitly in your report.

No

It is a condition of publication that authors make their supporting data, code and materials available - either as supplementary material or hosted in an external repository. Please rate, if applicable, the supporting data on the following criteria.

Is it accessible?

N/A

Is it clear?

N/A

Is it adequate?

N/A

Do you have any ethical concerns with this paper?

No

Comments to the Author

The concerns about the manuscript can be summed up with the question of who is the target audience?

If it is for people already aware of sedaDNA methods/field, is this reasoning/motivation in the review novel? It is certainly compelling, but the studies discussed also describe the same ideas. If the target audience are those capable of the work but not yet involved, then reasoning for why this may be of interested is provided, but the review isn't comprehensive enough (by itself) to suggest how. And if for those who may be interested but don't now where to start, there isn't enough detail in the methods (in this review) or nuance to suggest how to go about it. Clarifications about the goal(s) of the review can help address this concern.

Decision letter (RSPB-2021-1252.R0)

24-Jul-2021

Dear Dr Dalén

I am pleased to inform you that your manuscript RSPB-2021-1252 entitled "Integrating multi-taxon palaeogenomes and sedimentary ancient DNA to study past ecosystem dynamics" has been accepted for publication in Proceedings B.

The referees have recommended publication, but also suggest some minor revisions to your manuscript. As reviewer 2 notes, if you want to pull in an audience not already knowledgeable about the topic (and I hope you do...), you need to add some detail. For example, following referee 2's lead, if the listed complications/issues with sedaDNA work, what existing, validated or cutting edge methods exist to tackle them? How have previous studies handled these issues? What limitations exist in analyses, and how might future study design take these limitations into account? Therefore, with these points in mind, I invite you to respond to the referees' comments and revise your manuscript. Because the schedule for publication is very tight, it is a condition of publication that you submit the revised version of your manuscript within 7 days. If you do not think you will be able to meet this date please let us know.

Best wishes,
Innes Cuthill

Professor Innes Cuthill
Reviews Editor, Proceedings B
mailto: proceedingsb@royalsociety.org

Reviewer(s)' Comments to Author:

Referee: 1

Comments to the Author(s)

This is a timely review and prospectus of ancient DNA research applied to understanding ecosystem dynamics. In particular, the authors focus on paleogenomic and sedimentary metagenomics research and do an excellent job of summing up the major advances in the field as well as the challenges. I have only a few comments that might improve the manuscript. For example, I think that pointing to some of the potential ethical issues associated with sedaDNA is important. In the Americas, Australia and elsewhere, it is important to consult with indigenous stakeholders about research that might take place on their lands and to think carefully about the narrative that is used to explain the results of the research (see <https://doi.org/10.1038/s41559-020-01351-6>). Section b is one place where this could be mentioned, since in lines 199-203, the authors note that it would of interest to investigate the arrival of humans and the link to megafaunal extinction. Megafaunal extinction was likely a complex process where humans may or may not have been important players.but we need to avoid simplistic narratives leaving the general public with the conclusion that "X people killed off the Y animal(s)", particularly where

the descendents of “X people” are politically marginalized and when a more contextualized and complex answer is more likely to be true.

Additional comments:

Line 254: The following sentence could be clearer. “For instance, the introduction of the dingo in Australia likely contributed to the extinction of the thylacine and the Tasmanian devil on the mainland [85], whereas the use of hunting dogs led to a higher hunting success and increased pressure on ungulate populations (e.g., moose [68]).” Specifically, I was trying to figure out whether the dingo was a hunting dog (were they?)...and what moose were in Australia.

Referee: 2

Comments to the Author(s)

The concerns about the manuscript can be summed up with the question of who is the target audience?

If it is for people already aware of sedaDNA methods/field, is this reasoning/motivation in the review novel? It is certainly compelling, but the studies discussed also describe the same ideas. If the target audience are those capable of the work but not yet involved, then reasoning for why this may be of interested is provided, but the review isn't comprehensive enough (by itself) to suggest how. And if for those who may be interested but don't now where to start, there isn't enough detail in the methods (in this review) or nuance to suggest how to go about it. Clarifications about the goal(s) of the review can help address this concern.

Author's Response to Decision Letter for (RSPB-2021-1252.R0)

See Appendix A.

Decision letter (RSPB-2021-1252.R1)

02-Aug-2021

Dear Dr Dalén

I am pleased to inform you that your manuscript entitled "Integrating multi-taxon palaeogenomes and sedimentary ancient DNA to study past ecosystem dynamics" has been accepted for publication in Proceedings B.

If you are likely to be away from e-mail contact during this period, let us know. Due to rapid publication and an extremely tight schedule, if comments are not received, we may publish the paper as it stands.

Data Accessibility section

Open access

You are invited to opt for open access via our author pays publishing model. Payment of open access fees will enable your article to be made freely available via the Royal Society website as soon as it is ready for publication. For more information about open access publishing please visit our website at http://royalsocietypublishing.org/site/authors/open_access.xhtml.

The open access fee is £1,700 per article (plus VAT for authors within the EU). If you wish to opt for open access then please let us know as soon as possible.

Paper charges

Sincerely,

Proceedings B

Appendix A

Responses to Editor's and Reviewers' Comments to Author:

Editor:

E#1.1 As reviewer 2 notes, if you want to pull in an audience not already knowledgeable about the topic (and I hope you do...), you need to add some detail. For example, following referee 2's lead, if the listed complications/issues with sedaDNA work, what existing, validated or cutting edge methods exist to tackle them? How have previous studies handled these issues? What limitations exist in analyses, and how might future study design take these limitations into account?

Response to E#1.1: We thank the editor for these suggestions and fully agree that our review should target as broad of an audience as possible. With this in mind, we have expanded parts of section 3 and extensively expanded section 5a. We now better describe methods and challenges (for those unfamiliar with the field) and provide details on cutting edge methods, how these problems are currently being solved, and what new approaches are likely to be needed going forward (for the more experienced readers, including those currently within the *sedaDNA* field). We feel that these additions simultaneously target all three audience groups highlighted by Reviewer 2.

Referee: 1

Comments to the Author(s)

This is a timely review and prospectus of ancient DNA research applied to understanding ecosystem dynamics. In particular, the authors focus on paleogenomic and sedimentary metagenomics research and do an excellent job of summing up the major advances in the field as well as the challenges. I have only a few comments that might improve the manuscript.

We thank the Referee for their appreciation of our work.

R#1.1. For example, I think that pointing to some of the potential ethical issues associated with *sedaDNA* is important. In the Americas, Australia and elsewhere, it is important to consult with indigenous stakeholders about research that might take place on their lands and to think carefully about the narrative that is used to explain the results of the research (see https://url11.mailanyone.net/v1/?m=1m7F2I-0005op-5X&i=57e1b682&c=O74psAYSqIbr2Wkjl8S_QeRIH4bNBjxqS95aSyby-apcwtjRu4Tnhsi2VFbtSqSWk84r01bUN2XfAJwCBAdIn25xILSpmfgbvLSD9k060Rbn-BKfxCfMb4r9eIBkRH9evRC-HLuHzVBFESzr4GRvNyPuSC-cf-1cPyyhXDnh2JxTILis0c_yO0pZAUZfQeiH4U-wIo1rsI6OJndfE2dnptuyNnjINvRJu_XdM0ZcfxLJkBxkTUD2UGeYiLQFWDF). Section b is one place where this could be mentioned, since in lines 199-203, the authors note that it

would of interest to investigate the arrival of humans and the link to megafaunal extinction. Megafaunal extinction was likely a complex process where humans may or may not have been important players.....but we need to avoid simplistic narratives leaving the general public with the conclusion that “X people killed off the Y animal(s)”, particularly where the descendants of “X people” are politically marginalized and when a more contextualized and complex answer is more likely to be true.

Response to R#1.1: We thank Referee 1 for raising this very important point. We have now edited this section to emphasise the multifactorial causes of species extinction and the crucial need to engage with indigenous communities on L. 238-243: “Because megafaunal extinctions are often complex and multifactorial, a multi-taxon palaeogenomics approach will be especially valuable for assessment of the respective roles of human and non-human environmental changes in species extinction. In this regard, we stress that ethical considerations, engagement with indigenous communities, as well as careful interpretation of the narrative stemming from these discoveries will be essential to avoid any potential stigmatisation of indigenous peoples [75]”.

Additional comments:

R#1.2. Line 254: The following sentence could be clearer. “For instance, the introduction of the dingo in Australia likely contributed to the extinction of the thylacine and the Tasmanian devil on the mainland [85], whereas the use of hunting dogs led to a higher hunting success and increased pressure on ungulate populations (e.g., moose [68]).” Specifically, I was trying to figure out whether the dingo was a hunting dog (were they?)....and what moose were in Australia.

Response to R#1.2: We apologise for conflating these two independent points and have now split this sentence into two to clarify. We point out that dingos were used to assist humans in hunting in Australia. Hunting dogs were used on other continents to hunt ibex and gazelles (note that we changed the example taxa to give archaeological examples). The text has been updated on L. 294-298 to “For instance, since dingos were used to assist human hunting of small and large prey, their introduction in Australia likely contributed to the extinction of the thylacine and the Tasmanian devil on the mainland [89.90]. Similarly, the use of hunting dogs on other continents may have led to a higher hunting success and increased pressure on ungulate populations (e.g., ibex, gazelle [91]).”

Referee: 2

Comments to the Author(s)

R#2.1. The concerns about the manuscript can be summed up with the question of who is the target audience?

If it is for people already aware of sedaDNA methods/field, is this reasoning/motivation in the review novel? It is certainly compelling, but the studies discussed also describe the same ideas.

If the target audience are those capable of the work but not yet involved, then reasoning for why this may be of interested is provided, but the review isn't comprehensive enough (by itself) to suggest how. And if for those who may be interested but don't now where to start, there isn't enough detail in the methods (in this review) or nuance to suggest how to go about it. Clarifications about the goal(s) of the review can help address this concern.

Response to R#2.1: We thank the reviewer for highlighting gaps in the sedaDNA section and that its target audience was unclear. As we wish to target as broad an audience as possible, we have made changes to address each of the reviewer's three target audience groups. To maintain flow, we elected to keep the sedaDNA text split into two parts (overview: section 3, and issues/solutions/future challenges: section 5a), but have now expanded each of these.

For target group 1 ("those who may be interested but don't know where to start"), we have added descriptions for each of the three methods (metabarcoding, shotgun metagenomics, target enrichment) in section 3, to make the review more accessible, and have signposted to two recent detailed reviews. On L. 120-128, we now state: "*Subsequently, the majority of studies have used PCR-based DNA metabarcoding methods to amplify sedaDNA molecules of interest from individual broad taxonomic groups (e.g., plants or mammals; [40–42]). Advanced methods that sequence entire sedaDNA molecules, and thereby allow for aDNA damage authentication (see also section 5a), have only recently been applied. These methods include shotgun metagenomics, whereby any molecules in the sedaDNA mixture are randomly sequenced (e.g., [15,36,43–46]), and target enrichment, in which sedaDNA molecules of interest are selectively enriched prior to sequencing (e.g., barcode or mitochondrial loci; [16,47–49]). Detailed descriptions of these methods applied to sedaDNA have been recently reviewed elsewhere ([50,51]).*". We have also added descriptions to each of the challenges presented in section 5a to improve accessibility.

For group 1 and group 2 ("those capable of the work but not yet involved"), we are now more explicit about how issues surrounding sedaDNA have been, or are being, solved in section 5a. For groups 2 and 3 ("people already aware of sedaDNA methods/field"), we have also added ongoing and potential future analytical challenges and solutions, the latter of which are novel to the best of our knowledge. We have extensively revised and expanded section 5a on L. 353-392.

Lastly, to reflect these changes and the review as a whole, we have modified the review goal statement in the Abstract to include 'future prospects' on L. 40: "*In this review, we discuss the various applications, associated challenges, and future prospects of such an approach.*".